# Developing the Entrepreneurial University: Factors of Influence

**María José Bezanilla \***, **Ana García-Olalla, Jessica Paños-Castro** and **Arantza Arruti**

Faculty of Psychology and Education, University of Deusto, 48007 Bilbao,
Spain; ana.garciaolalla@deusto.es (A.G.-O.); jessicapanos@deusto.es (J.P.-C.); aarruti@deusto.es (A.A.)
**\*** Correspondence: marijose.bezanilla@deusto.es; Tel.: +34-94-413-90-03

**Abstract:** Universities are increasingly paying more attention to developing academic entrepreneurship. This paper analyses the existing relationships between the relevant factors that contribute to the development of the entrepreneurial university. A previously validated questionnaire was administered to a sample of 84 deans of a number of faculties in Spain. The aim was to assess the universities' development in terms of 13 influencing factors in encouraging entrepreneurship. The findings show that universities' contextual factors had only minor influence on internal factors. Internal resources were found to be moderately or highly correlated with the processes put in place by universities to promote entrepreneurship. In particular, reference to entrepreneurship in a university's mission, strategy, policies and procedures had a correlation with all the entrepreneurship factors analysed. Support from the management team and organisational design were not among the most important factors; however, they were positively associated with training and research processes, which, in turn, seemed to be strongly related to all factors in the development of the entrepreneurial university, especially with university mission and strategy. The findings show the relationships between the factors involved in the development of the entrepreneurial university. This will help universities to adopt measures that are better suited to promoting entrepreneurship.

**Keywords:** entrepreneurial university; entrepreneurship; triple helix; higher education; influencing factors

## 1. Introduction

Universities emerged from monastic (scholastic) schools in the twelfth and thirteenth centuries. Their teaching and research roles were only added much later with the creation of the University of Berlin, which has been regarded as the first modern university, and which was created in 1810 [1]. Currently, five types of universities can be identified in terms of their role [2]. First, the academic university, which is largely aimed at teaching students. Second, the classic university, where research is combined with teaching. Third, the social university, which takes an active part in the discussion and resolution of society's problems. Fourth, the business university, where teaching and Research and Development activities are carried out based on business criteria. Fifth, the entrepreneurial university, which has a strong role in the social context within which it operates. In addition to the basic teaching and research functions, a third key mission for society should be incorporated into universities: Fostering entrepreneurial projects or conducting development projects working together with other agents within the regional system [3]. Universities can be actively engaged in these projects, as they are close to the markets and have sound knowledge of the different trends as they emerge [4]. In today's knowledge society, universities are increasingly and more directly becoming promoters of economic and social development [2]. Universities have recognized the role of education in building societies based on values of equity, social justice and sustainability.

The term entrepreneurial university was coined by Etzkowitz in 1998 [5], and refers to regional economic development [6]. This concept is also known as the triple helix model, which describes the relationship between universities, industries and governmental organisations intended to stimulate innovation [7], create incubators and/or support structures for lecturers and students to start new businesses [8] and raise awareness of and promote entrepreneurship [9]. Other authors, including Subotzky (1999, quoted in 8), have defined entrepreneurial universities as those where a closer partnership exists between academia and businesses; where faculties have greater responsibility for obtaining external funding; and where there is a managerial ethos in institutional governance, leadership and planning. It should be noted that the relationship between these three actors, university, industry and government, is interdependent; in other words, these actors condition each other and constitute an organic unit [10].

While no consensus exists about a single definition of the entrepreneurial university, several authors have listed a number of features that characterise it [11]. However, there are few models that explain the entrepreneurial university's foundations and conceptual basis. There is also a paucity of empirical studies on the subject [7]. The majority of the research carried out has been based on conceptual frameworks that seek to identify the features that should characterise the entrepreneurial university. As an example, O'Shea et al. [6] proposed a number of factors that could bolster the entrepreneurial university, namely top-down leadership, policies that support and encourage the process of academic entrepreneurship, own funding, technological transfer offices and incubators, an entrepreneurial culture, entrepreneurial attitudes and aptitudes, access to venture capital, infrastructures and technology clusters. Baporikar [12] highlights the context and organizational aspects to understand the entrepreneurial university.

Only a few studies have empirically analysed the entrepreneurial university [8,13–16], most of which have relied on the different factors involved in building their conceptual frameworks. Moreover, Riviezzo, Napolitano and Fusco [17] highlight that empirical assessment of the social and cultural impacts of the university in a community has been largely overlooked.

A specific area of research interest is in analysing factors based on the creation of university spin-offs, which seeks to identify why some universities are more successful at generating them [6,18–22]. However, the literature is scarce on the factors that make up the entrepreneurial university as a whole; instead, research tends to be focused on proposing theoretical models that are yet to be empirically demonstrated [23,24].

Based on their review of 17 studies centred on identifying which factors are important in fostering the entrepreneurial university, and how the entrepreneurial university influences regional development, Guerrero-Cano et al. [25] indicated three formal factors (governance organisation and structure; support measures to start new businesses; and an entrepreneurial university) and three informal factors (attitudes of the university community; teaching methodologies aimed at entrepreneurship; and an academic reward system) that are conducive to strengthening the entrepreneurial university.

In 2013, the Organisation for Economic Co-Operation and Development (OECD, 2013) and the European Commission issued the self-evaluation tool HEInnovate (higher education institutions innovation) [26]. This tool is intended to assess the entrepreneurial and innovative potential of higher education institutions across eight key areas. The key areas directly related to the aims of this paper are those related to entrepreneurial teaching and learning, and to the support of entrepreneurs.

Since the studies published to date are not focused on models that encompass all the factors covered in the literature, there is a need to further the knowledge of the determinant factors of the entrepreneurial university, and the extent of their influence. To this end, Errasti et al. [27] devised and validated a model of maturity for the measurement of the level of academic entrepreneurship among faculties and universities. The model included thirteen factors: Legal and administrative context; business and organisational context; entrepreneurship funding; training in entrepreneurship for faculty staff; inclusion of professionals from businesses and organisations in the development and delivery of the curriculum; mission and strategy; policies and procedures; support from the management team;

organisational design; training and research in entrepreneurship; extra-curricular training; active methodologies; and internationalisation.

The study conducted by Errasti et al. [27] concluded that there was a modest degree of development in the various elements involved in the entrepreneurial university, and that there is still much room for improvement. It also showed that the most developed factors among Spanish universities were internationalisation, use of active methodologies, mission and strategy and support from the management team. In contrast, the least developed factors were found to be entrepreneurship funding; business and organisational context; training in entrepreneurship for faculty staff; and the legal and administrative context.

While some studies have focused on identifying which factors promote the entrepreneurial university, others have highlighted some potential correlations between those factors. This is the case of Fini et al. [21], who held that a close interaction between local businesses or organisations and the university help create a social environment that supports people and encourages them to share knowledge and ideas. Guerrero and Urbano [28] and Hu [29], for their part, argued that funding was essential for a university's autonomy and development. Davey et al. [30] and De Luca et al. [31] concurred that the collaboration of external business experts in advising on and developing the curriculum fosters both university-business cooperation processes and the acquisition of key skills by future entrepreneurs.

Errasti et al. [27] presented a descriptive analysis of the factors that contribute to the development of the entrepreneurial university. Following this study, the main research question of the present paper is: What are the existing relations among these factors? Therefore, the overall aim of this study is to determine whether there are associations between the levels of development of some factors involved in the entrepreneurial university that indicate significant influence relations to help further its advancement. This aim will be specifically focused on the following objectives:

1. To identify to what extent external contextual factors are related to the development of the entrepreneurial university.
2. To identify to what extent having different types of resources relates to the development of the entrepreneurial university.
3. To analyse to what extent the processes that a university puts in place in connection with its projects, structure and training are interlinked and also connected to other external and internal factors.

## 2. Hypotheses and Variables

As indicated, the variables under study in this work have been the characteristics of the entrepreneurial university identified by the literature and that have been previously validated as constituent factors of it. More specifically, in order to respond to the objectives set out, the relationship between these variables has been analysed. To this end, the hypotheses expressing the expected relationships between these factors have been formulated, based on the literature and previous research. Below, we detail how the conceptual definition of these variables has been made, as well as the operational definition through the elements of each one of them that are evaluated with the instrument developed for their measurement.

The hypothesis that guide the study have been organised according to the CIPP model [32], a model for institutional evaluation that uses contextual, input, process and product factors. The hypothesis of the study can be stated as follows.

In the first place, the models for institutional evaluation that use contextual, process and product factors have traditionally advocated that contextual factors help to explain other internal (process and product) factors and variables, since they have an impact on them to a greater or lesser extent. This impact has been attributed to the fact that they outline the conditions for intervention, as they consider explanatory and control variables related to political, legal, administrative, demographic, socio-economic and cultural conditions.

**Hypothesis 1 (H1).** *External contextual factors (Legal and Administrative Context, Business and Organizational Context) are positively related to the development of internal factors. This hypothesis has been supported by previous literature on the development of the entrepreneurial university* [33–35]. See Table 1.

**Table 1.** Hypotheses and variables (conceptual and operational definition) related to the context of the entrepreneurial university.

| Hypotheses | Variables—Conceptual Definition of Factors | | Elements for Its Operational Definition | | Alpha |
|---|---|---|---|---|---|
| | 1. | Legal and administrative context: Government and public administrations become involved in and facilitate entrepreneurship. | 1.1. 1.2. 1.3. | Legislation Financing  Public infrastructures | 0.896 |
| Hypothesis 1 (H1) | 2. | Business and organisational context: Nearby organisations and companies that operate in the same or a similar business sector, and interact with the university by sharing the same field of research, knowledge and ideas through formal and informal networks. | 2.1. 2.2. 2.3. 2.4.  2.5. | Financing Technological level Innovation level  Technological maturity R & D budget | 0.893 |

Secondly, resource factors are intended to account for the inflows into the system, both in the form of material (economic, infrastructure) resources and human resources (staff) available for an organisation to operate. See Table 2.

**Hypothesis 2 (H2).** *Resource factors (Entrepreneurship funding, Training in Entrepreneurship for Faculty Staff, Inclusion of professionals) are related to the development of institutional statements linked to entrepreneurship (Mission and Strategy, Policies and Procedures).*

**Hypothesis 3 (H3).** *Resource factors (Entrepreneurship funding, Training in Entrepreneurship for Faculty Staff, Inclusion of professionals) are associated with the development of structures that support entrepreneurship (Support from the management team, Organisational Design).*

**Hypothesis 4 (H4).** *Resource factors (Entrepreneurship funding, Training in Entrepreneurship for Faculty Staff, Inclusion of professionals) are related to the development of processes for entrepreneurship (Training and Research in entrepreneurship, Extra-curricular entrepreneurship training, Active methodologies, Internationalisation).*

**Table 2.** Hypotheses and variables (conceptual and operational definition) related to the resources of the entrepreneurial university.

| Hypotheses | Variables—Conceptual Definition of Factors | Elements for Its Operational Definition | Alpha |
|---|---|---|---|
| Hypothesis 2 (H2) Hypothesis 3 (H3) Hypothesis 4 (H4) | 3. Entrepreneurship funding: This factor demonstrates the autonomy of the university, shows the funds for research and teaching in entrepreneurship, and for creating entrepreneurship projects and setting up companies and organisations. | 3.1. Funding for entrepreneurship teaching 3.2. Funding for research into entrepreneurship 3.3. Seed Capital | 0.880 |
| | 4. Training in entrepreneurship for faculty staff: Extent to which the university provides training in entrepreneurship to its staff, in terms of transfer of knowledge and in the creation of spin-offs so that they can promote entrepreneurship among their students. | 4.1. Training in entrepreneurship 4.2. Transfer of knowledge 4.3. Creation of spin-offs | 0.878 |
| | 5. Inclusion of professionals from businesses and organisations in the development and delivery of the curriculum: Examines the presence of experts from the business world and/or practising professionals or agents from nearby organisations and/or from the same business sector in the design, development and delivery of the curriculum. Includes university-company collaboration in the development of course programmes, modules, experiences, etc., as well as the inclusion of guest lecturers. | 5.1. Participation in the main governing body of the faculty 5.2. Participation in development and delivery 5.3. Lecturers and guest professionals | 0.889 |

In the third place, process-related factors account for the processes that an organisation puts in place to operate and provide its services. They may be related to the projects it carries out, which guide its actions; to the structures organised to implement them; or to the training processes whereby it operates (by means of the key training process). See Table 3.

**Table 3.** Hypotheses and variables (conceptual and operational definition) related to the processes—projects, structures, training and research—developed by the entrepreneurial university.

| Hypotheses | Variables—Conceptual Definition of Factors | Elements for Its Operational Definition | Alpha |
|---|---|---|---|
| Hypothesis 5 (H5) Hypothesis 6 (H6) Hypothesis 7 (H7) | 6. Mission and strategy: Analyses whether the mission statement and strategies of the university include the word 'Company/Organisation' or 'Entrepreneurship' in any of its documents (mission, vision values, strategic plan). | 6.1. Presence in the mission<br>6.2. Objectives<br>6.3. Strategy on knowledge transfer<br>6.4. Strategy for university-business/organisation partnership<br>6.5. Strategy for entrepreneurship<br>6.6. Strategies related to social responsibility<br>6.7. Monitoring and evaluation of results | 0.876 |
|  | 7. Policies and procedures: Evaluates the existence and possible influence of university policies, procedures and practices on Academic Entrepreneurship Activities, such as university policies on intellectual property and networking activities for university-business collaboration, and university spin-offs. | 7.1 Policies and procedures on knowledge transfer<br>7.2. Policies and procedures for university-business/organisation partnership<br>7.3. Policies and procedures for the creation of spin-offs | 0.879 |
|  | 8. Support from the management team: Analyses the leadership, understanding and support of the management team regarding the entrepreneurial culture in the university, as shown in decision making, behaviours and actions that influence the university's strategy | 8.1 Support for entrepreneurship<br>8.2. Revenue for entrepreneurship<br>8.3. Presence on the agenda | 0.887 |

**Table 3.** *Cont.*

| Hypotheses | Variables—Conceptual Definition of Factors | Elements for Its Operational Definition | Alpha |
|---|---|---|---|
| | **9.** Organisational design: Analyses the extent to which a university facilitates entrepreneurial behaviour within it through its own organisational design mechanisms, such as the decentralisation of decision making, flexibility in the integration of strategies, financial autonomy, the relationship between teaching and research and the degree to which individuals have the power to innovate. | 9.1　Connection between teaching and research<br>9.2.　Decentralised decision-making<br>9.3.　Bottom-up structure<br>9.4.　Financial autonomy | 0.890 |
| | **10.** Training and research in entrepreneurship: Formal education in entrepreneurship can be defined as the development of competences (behaviours, knowledge, skills and attitudes) specific to the person within academic curricula and in research. | 10.1.　Entrepreneurial skills in the curriculum<br>10.2.　Specific programmes on entrepreneurship<br>10.3.　Research | 0.876 |
| | **11.** Extra-curricular training: The extra-curricular training process for academic entrepreneurship refers to the training activities carried out outside the curriculum, such as awareness-raising, workshops for the identification of opportunities and courses for the implementation of innovative projects, the development of business plans and the launch of spin-offs. | 11.1.　Raising awareness about entrepreneurship<br>11.2.　Identification of opportunities<br>11.3.　Business plan development<br>11.4.　Launch of spin-offs | 0.878 |

**Table 3.** *Cont.*

| Hypotheses | Variables—Conceptual Definition of Factors | Elements for Its Operational Definition | Alpha |
|---|---|---|---|
| | 12. Active methodologies: Entrepreneurship education professionals should be able to create an open environment in which students develop the confidence to take risks and learn from their successes and failures, participation in real projects and works, are all active methodologies that can foster the development of entrepreneurship. | 12.1. Use of active methodologies 12.2. Placements with entrepreneurs 12.3. Design and development of innovative educational resources | 0.885 |
| | 13. Internationalisation: Development of joint degrees with universities abroad, the carrying out of international research projects and the mobility activities of students, academics and/or partners are key elements of the entrepreneurial university | 13.1 Joint degrees 13.2. Research 13.3. Revenues 13.4. Mobility | 0.892 |

**Hypothesis 5 (H5).** *The existence of entrepreneurship Projects (Mission and Strategy, Policies and Procedures) would be related to Structures that support entrepreneurship (Support from the management team, Organisational design).*

**Hypothesis 6 (H6).** *The existence of entrepreneurship Projects (Mission and Strategy, Policies and Procedures) would be associated with entrepreneurship Training Processes (Formal entrepreneurship training, Extra-curricular entrepreneurship training, Active methodologies, Internationalisation).*

**Hypothesis 7 (H7).** *The existence of Structures that support entrepreneurship (Support from the management team, Organisational design) could be expected to be related to the development of entrepreneurship Training Processes (Formal entrepreneurship training, Extra-curricular entrepreneurship training, Active methodologies, Internationalisation).*

## 3. Materials and Methods

The sample invited to participate in this study was made up of 567 faculties, schools and affiliated centres belonging to 44 universities (public and private) from six autonomous regions, those identified in the literature as being benchmarks in academic entrepreneurship in Spain [36,37]. Participation was voluntary, and confidentiality was guaranteed through a letter requesting their cooperation and informed consent.

A total of 98 subjects from the invited sample responded to the request for participation. After a preliminary analysis, 14 were eliminated because of their atypical responses (extreme and outstanding cases in the box diagram). This led to the final sample consisting of 84 cases, 14.81% of the invited sample.

The percentage of participation of public institutions was greater than that from private ones (76% and 24%, respectively). Five different areas were taken into account for the study. While the subject areas were not homogeneously represented, the participating autonomous regions were.

In order to meet the research objectives, a questionnaire was used that had been previously designed and validated by Errasti et al. [27] to measure the maturity of academic entrepreneurship among different faculties. This questionnaire, based on the original instrument by Markuerkiaga et al. [38], consisted of 14 blocks of mostly closed questions, with the inclusion of a smaller number of open questions to allow participants to provide evidence and/or add comments and clarifications. The first 13 blocks were required to be answered, while block 14 was optional. The questionnaire was preceded by a section where each faculty's general and descriptive data were recorded.

The questionnaire included a total of 13 blocks and 48 elements, with mandatory matrix questions rated on a 10-point Likert-type scale, and were grouped into three levels (low, medium and high). A rubric with descriptors and mutually exclusive response options were employed. The blocks and elements included have already been stated in the previous section. This dimension also contained an optional open-ended question aimed at obtaining comments, clarifications and evidence that the subjects may wish to provide.

Following the design of the contacts database, the questionnaire was sent from the internal messaging system of the Qualtrics programme (tool used for creating, collecting and consolidating surveys). A reminder was sent two weeks later and another one within a month after the questionnaire had been distributed. All data collection was conducted online.

It is important to emphasise that, during the data collection process, all necessary steps were taken to ensure the confidentiality of the participants. Time and resources were devoted to explaining the purpose and nature of the research. The individual freedom to participate in the research was respected at all times, and participants were informed about how the results would be used [39].

## 4. Results

As the variables used were quantitative, the correlation index that expressed 'an estimation of the degree to which two variables vary together' was analysed, in order to study the relationship between the thirteen factors assessed by the questionnaire [40]. Pearson's correlation coefficient was employed in order to analyse the relations between variables, considering those associations that were significant at a confidence level of 0.99 and 0.95. Moreover, the size of the correlation was valued as low, moderate or high, in accordance with the recommendations made by Bisquerra [39] in the field of educational sciences. In light of the wide range of the variables, the most important findings will be discussed in connection to the different types of factors. Table 4 shows the factors correlation matrix.

### 4.1. Contextual Factors

Table 4 shows the correlations between Factor 1, legal and administrative context, and Factor 2, business and organizational context, with the rest of the factors. These conclusions can be drawn from the data analysis:

- Factor 1, legal and administrative context, showed a significant but low correlation with the following factors:

  - Resource factors: Funding for entrepreneurship (0.21, significant at the 0.05 level) and Training in entrepreneurship for faculty staff (0.29, significant at the 0.01 level).
  - Project-related factors: Mission and strategy (0.24, significant at the 0.05 level) and policies and procedures (0.32, significant at the 0.05 level).
  - Structural factors: Organisational design (0.22, significant at the 0.05 level).
  - Training process factors: Training and research in entrepreneurship (0.25, significant at the 0.05 level) and extra-curricular training (0.27, significant at the 0.05 level).

- Factor 2, business and organizational context, showed a low significant correlation with the following factors:

  - Resource factors, funding for entrepreneurship (0.24, significant at the 0.05 level) and training in entrepreneurship for faculty staff (0.29, significant at the 0.01 level).
  - Project-related factors: Mission and strategy (0.24, significant at the 0.05 level) and policies and procedures (0.27, significant at the 0.05 level).
  - Structural factors: Organisational design (0.22, significant at the 0.05 level).
  - Training process factors: Training and research in entrepreneurship (0.23, significant at the 0.05 level), extra-curricular training (0.30, significant at the 0.05 level) and internationalisation (0.26, significant at the 0.05 level).

To further the analysis. Student's t-test was used to compare those faculties that were above and below the mean score of these two factors. No significant difference was found between them in terms of their level of development of the remaining factors.

**Table 4.** Factors correlation matrix.

| | F1 | F2 | F3 | F4 | F5 | F6 | F7 | F8 | F9 | F10 | F11 | F12 | F13 |
|---|---|---|---|---|---|---|---|---|---|---|---|---|---|
| F1.Legal and administrative context | 1 | 0.627 ** | 0.218 * | 0.293 ** | 0.156 | 0.244 * | 0.325 ** | −0.003 | 0.223 * | 0.256 * | 0.276 * | 0.154 | 0.205 |
| F2.Business and organisational context | 0.627 ** | 1 | 0.247 * | 0.290 ** | 0.214 | 0.240 * | 0.274 * | 0.139 | 0.220 * | 0.237 * | 0.301 ** | 0.013 | 0.267 * |
| F3.Entrepreneurship funding | 0.218 * | 0.247 * | 1 | 0.605 ** | 0.453 ** | 0.548 ** | 0.423 ** | 0.503 ** | 0.322 ** | 0.596 ** | 0.571 ** | 0.555 ** | 0.275 * |
| F4.Training in entrepreneurship for faculty staff | 0.293 * * | 0.290 ** | 0.605 ** | 1 | 0.356 ** | 0.595 ** | 0.640 ** | 0.362 ** | 0.294 ** | 0.599 ** | 0.670 ** | 0.440 ** | 0.424 ** |
| F5.Inclusion of professionals | 0.156 | 0.214 | 0.453 ** | 0.356 ** | 1 | 0.457 ** | 0.407 ** | 0.384 ** | 0.276 * | 0.407 ** | 0.358 ** | 0.471 ** | 0.124 |
| F6.Mission and strategy | 0.244 * | 0.240 * | 0.548 ** | 0.595 ** | 0.457 ** | 1 | 0.695 ** | 0.508 ** | 0.448 ** | 0.775 ** | 0.611 ** | 0.549 ** | 0.448 ** |
| F7.Policies and procedures | 0.325 ** | 0.274 * | 0.423 ** | 0.640 ** | 0.407 ** | 0.695 ** | 1 | 0.391 ** | 0.465 ** | 0.521 ** | 0.529 ** | 0.393 ** | 0.421 ** |
| F8.Support from the management team | −0.003 | 0.139 | 0.503 ** | 0.362 ** | 0.384 ** | 0.508 ** | 0.391 ** | 1 | 0.314 ** | 0.510 ** | 0.475 ** | 0.369 ** | 0.306 ** |
| F9.Organisational design | 0.223 * | 0.220 * | 0.322 ** | 0.294 ** | 0.276 * | 0.448 ** | 0.465 ** | 0.314 ** | 1 | 0.348 ** | 0.196 | 0.317 ** | 0.349 ** |
| F10.Training and research in entrepreneurship | 0.256 * | 0.237 * | 0.596 ** | 0.599 ** | 0.407 ** | 0.775 ** | 0.521 ** | 0.510 ** | 0.348 ** | 1 | 0.694 ** | 0.533 ** | 0.411 ** |
| F11.Extra-curricular training | 0.276 * | 0.301 ** | 0.571 ** | 0.670 ** | 0.358 ** | 0.611 ** | 0.529 ** | 0.475 ** | 0.196 | 0.694 ** | 1 | 0.580 ** | 0.351 ** |
| F12.Active methodologies | 0.154 | 0.013 | 0.555 ** | 0.440 ** | 0.471 ** | 0.549 ** | 0.393 ** | 0.369 ** | 0.317 ** | 0.533 ** | 0.580 ** | 1 | 0.214 |
| F13.Internationalisation | 0.205 | 0.267 * | 0.275 * | 0.424 ** | 0.124 | 0.448 ** | 0.421 ** | 0.306 ** | 0.349 ** | 0.411 ** | 0.351 ** | 0.214 | 1 |

** Correlation is significant at level 0.01 (bilateral). * Correlation is significant at level 0.01 (bilateral).

Based on these data, it can be concluded that the hypothesis H1 that external factors would be associated with and influence the development of internal factors has not been validated, except for the training in entrepreneurship for staff. At first sight, this seems to contradict the hypothesis that was proven in previous studies. However, taking into account the low scores obtained for all factors and variables in the context of this study, it may indicate that it does not have a significant influence on the sample of Spanish universities analysed due to the low degree of support given by them to entrepreneurship. As Spanish universities organise their entrepreneurship activities by relying on their own resources, their entrepreneurial development might be greater if the context were more favourable and supportive, as has been the case in other countries. Only if contextual factors obtained a high score and if this score were not correlated with the development of internal factors could it be stated that such influence between factors does not exist.

*4.2. Resource Factors*

Table 4 shows the correlations of these resource factors with each of the process-related factors. The findings were as follows:

- Factor 3 (entrepreneurship funding) was found to have a significant but moderate correlation with the majority of the process-related factors; specifically:

  - A moderate correlation with the two project-related factors: Mission and strategy (0.54, significant at the 0.01 level) and policies and procedures (0.42, significant at the 0.01 level);
  - A moderate and low correlation, respectively, with the two structural factors: Support from the management team (moderate, with 0.50, significant at the 0.01 level) and organisational design (low, with 0.32, significant at the 0.01 level); and
  - A moderate correlation with three of the training process factors: Training and research in entrepreneurship (0.59, significant at the 0.01 level); extra-curricular training (0.57, significant at the 0.01 level); and active methodologies (0.55, significant at the 0.01 level); and a low correlation with the other training process factor, internationalisation (0.27, significant at the 0.05 level).

- Factor 4, training in entrepreneurship for faculty staff, was found to have a significant (high or moderate) correlation with the majority of the process-related factors:

  - A moderate and high correlation with the two project-related factors: Moderate with mission and Strategy (0.59, significant at the 0.01 level) and high with policies and procedures (0.64, significant at the 0.01 level);
  - A low correlation with the two structural factors: Support from the management team (0.36, significant at the 0.01 level) and organisational design (0.29, significant at the 0.01 level); and
  - A moderate correlation with three training process factors and high with one: High with extra-curricular training (0.67, significant at the 0.01 level); and moderate with training and research in entrepreneurship (0.59, significant at the 0.01 level); active methodologies (0.44, significant at the 0.01 level); and internationalisation (0.42, significant at the 0.01 level).

- Factor 5, inclusion of professionals into the curriculum, had a lower correlation with process, as it was found to be either moderate or low:

  - A moderate correlation was seen with the two project-related factors: Mission and strategy (0.45, significant at the 0.01 level) and policies and procedures (0.40, significant at the 0.01 level).
  - A low correlation was seen with the two structural factors: Support from the management team (0.38, significant at the 0.01 level) and organisational design (0.27, significant at the 0.05 level).

- Correlation was moderate with two training process factors, and low with one of them, whereas there was no correlation with the other: A moderate correlation was found with training and research in entrepreneurship (0.40, significant at the 0.01 level) and active methodologies (0.47, significant at the 0.01 level); a low correlation was found with extra-curricular training (0.67, significant at the 0.01 level); and no correlation was seen for internationalisation.

To sum up, internal resource factors seemed to have a weak association with external contextual factors (against what might have been expected in H1); however, they were found to be moderately significantly associated with the development of internal processes for entrepreneurship. The strongest relationship and possible influence was found to be with training in entrepreneurship for faculty staff, and the strongest association was seen with the development of projects and with some training processes, confirming H3. Association seemed moderate (confirming partially H2) with funding for entrepreneurship. Inclusion of professionals seemed to have less association (no confirming H4); and the weakest association and possible effect was seen for the development of structures and the internationalisation process.

### 4.3. Process-Related Factors

#### 4.3.1. Project-Related Factors

Table 4 shows the correlations between project-related factors and the other factors.

- Let us remember correlations found between these two project-related factors and the previous factors: Low but significant correlation with the two contextual factors, and a moderate correlation with the three resource factors, which was high between training in entrepreneurship for faculty staff and politics and procedures.
- Factor 6, mission and strategy, was found to have a significant, moderate-to-high relationship with other process-related factors:

  - Moderate with structural factors: Support from the management team (0.50, significant at the 0.01 level) and organisational design (0.44, significant at the 0.01 level).
  - Correlation tended to be high with training process factors: Training and research in entrepreneurship (high, 0.77, significant at the 0.01 level); extra-curricular training (high, 0.61, significant at the 0.01 level); active methodologies (moderate, 0.54, significant at the 0.01 level); and Internationalisation (moderate, 0.44, significant at the 0.01 level).

- Factor 7 Policies and procedures tended to be less assocaited, with a significant but moderate relationship:

  - A low correlation was observed between structures and support from the management team (0.39, significant at the 0.01 level) while a moderate correlation was found with organisational design (0.46, significant at the 0.01 level).
  - Correlation was seen to be moderate with training process factors: Training and research in entrepreneurship (moderate, 0.52, significant at the 0.01 level); extra-curricular training (moderate, 0.52, significant at the 0.01 level); active methodologies (low, 0.39, significant at the 0.01 level); and internationalisation (moderate, 0.42, significant at the 0.01 level).

In summary, all project-related factors proved to be significantly associated with all the factors studied here, albeit to varying degrees. Project-related factors seemed to be minimally influenced by internal resources. However, they were found to be strongly associated with the development of the remaining internal processes for entrepreneurship, particularly with mission and strategy, which showed a high correlation with training processes (confirming H6). The two project-related

factors were also seen to be associated with the development of structures that support entrepreneurship (confirming H5), although moderately and to a lesser extent. These findings supported and validated the hypothesis that had been proved by previous studies [41] as to the importance that mission and strategy have in developing entrepreneurship at university.

### 4.3.2. Structure-Related Factors

Table 4 shows the correlations that structural factors had with other factors. Recalling briefly the relationships found with the factors previously analysed:

- A low correlation or no correlation with the external contextual factors.
- A low correlation with the three resource factors, which was only moderate between entrepreneurship funding and support from the management team.
- A tendency towards a moderate correlation with the two project-related factors, especially with mission and strategy.

- Factor 8, support from the management team, showed a moderate or low relationship with training processes:

  - A moderate correlation with training and research in entrepreneurship (0.51, significant at the 0.01 level) and with extra-curricular training (0.47, significant at the 0.01 level).
  - A low correlation with active methodologies (0.36, significant at the 0.01 level) and with internationalisation (0.30, significant at the 0.01 level).

- Factor 9, organisational design, had a significant relationship with training processes, although it tended to be low:

  - A low correlation with training and research in entrepreneurship (0.34, significant at the 0.01 level), active methodologies (0.31, significant at the 0.01 level) and internationalisation (0.34, significant at the 0.01 level).
  - No correlation was found with extra-curricular training.

These data seemed to reveal that Structural factors were not the most important in promoting the development of the entrepreneurial university (no confirming H7). They appeared to be minimally influenced by factors related to context and internal resources, except for the funding available. The most significant relationship was found between the Support from the management team and the training processes for entrepreneurship (curricular, extra-curricular and research processes).

### 4.3.3. Training-Related Factors

Table 4 shows the correlations between training-related factors and the other factors. Let us first recall the relationships identified with the factors analysed previously:

- They showed a low correlation or no correlation with external contextual factors.
- These factors had a tendency to be moderately correlated with resource factors. Correlation was only high between training for faculty staff and extra-curricular training.
- A significant correlation was identified with the two project-related factors, particularly with mission and strategy (with a tendency to have a high correlation). It was more moderate with policies and procedures.
- A moderate-to-low correlation was found with the structural factors. It was higher with support from the management team and curricular and extra-curricular training.

- In addition, a significant correlation was observed between the four factors included in the Training-related processes:

- A high correlation was identified for F10, training and research in entrepreneurship and F11, extra-curricular training for entrepreneurship (0.69, significant at 0.01), and a moderate correlation was found between F10 and the other factors F12, active methodologies (0.53, significant at the 0.01 level) and F13, internationalisation (0.41, significant at the 0.01 level).
- A moderate correlation was seen between F11, extra-curricular training for entrepreneurship and F12, active methodologies (0.58, significant at the 0.01 level).
- A low correlation was found between F11, extra-curricular training and F13 internationalisation (0.35, significant at the 0.01 level).
- No correlation was found between F12, active methodologies and F13 Internationalisation.

In short, Training-related factors showed the highest and most numerous correlations among all the factors. Despite this, they revealed low or no correlations with external contextual factors. In contrast, these factors were found to have the highest correlations with resource-related factors, which were lower and less frequent for other factors; and with structural factors, the most significant being the relationships between training for faculty staff and support from the management team with both curricular and extra-curricular training. Both of these tended to have the highest correlation with the other factors and therefore were the most sensitive to the influence of the remaining factors.

## 5. Discussion and Conclusions

As a conclusion, in the following figure, Figure 1, the main relations between factors found in the research can be seen. Factors have been grouped in external/contextual factors, resources and process factors, this last one including projects, structures and training processes. Factors in bold have been found as most associated to other factors of the model. These are: Entrepreneurship funding, training in entrepreneruship for faculty staff, mission and strategy, support from the management team, training and research in entrepreneurship and extra-curricular training. Arrows in bold indicate the most relevant relationships.

As it can be observed in Figure 1, external/context factors are mainly associated with training in entrepreneurship for faculty staff. Resources factors are mainly associated bilaterally to projects and training processes. Projects are associated bilaterally to structures and also training processes. It could be highlighted that training processes factors show the highest and most numerous correlations among all the factors, except with external/context factors. Those factors were found to have the highest correlations with resources factors and with projects factors.

The findings of the study showed a weak relationship between context and the development of the entrepreneurial university. The minimal influence of context was an unexpected result, since numerous studies had found precisely the opposite, namely that there was a strong relationship between the legal, administrative and economic context within which a university was placed and its entrepreneurial development [33–35]. This finding may be explained by the low scores obtained by the contextual variables in this study, or by universities relying on their own resources for entrepreneurial endeavours, thus demonstrating their autonomy. The legal and business context has traditionally had a very low level of development in Spain, and the tendency to greater awareness and support of entrepreneurship is only recent. This has been reflected in Spanish Law 14/2013, on support for entrepreneurs and their internationalisation [42], and others passed at a regional level, including Law 3/2018, on the promotion of Entrepreneurship in Andalusia and Law 16/2012, on support for entrepreneurs and small-sized enterprises in the Basque Country. It seems necessary to explore this aspect further; particularly, it would be interesting to assess the impact of this new framework in the medium- and long-term.

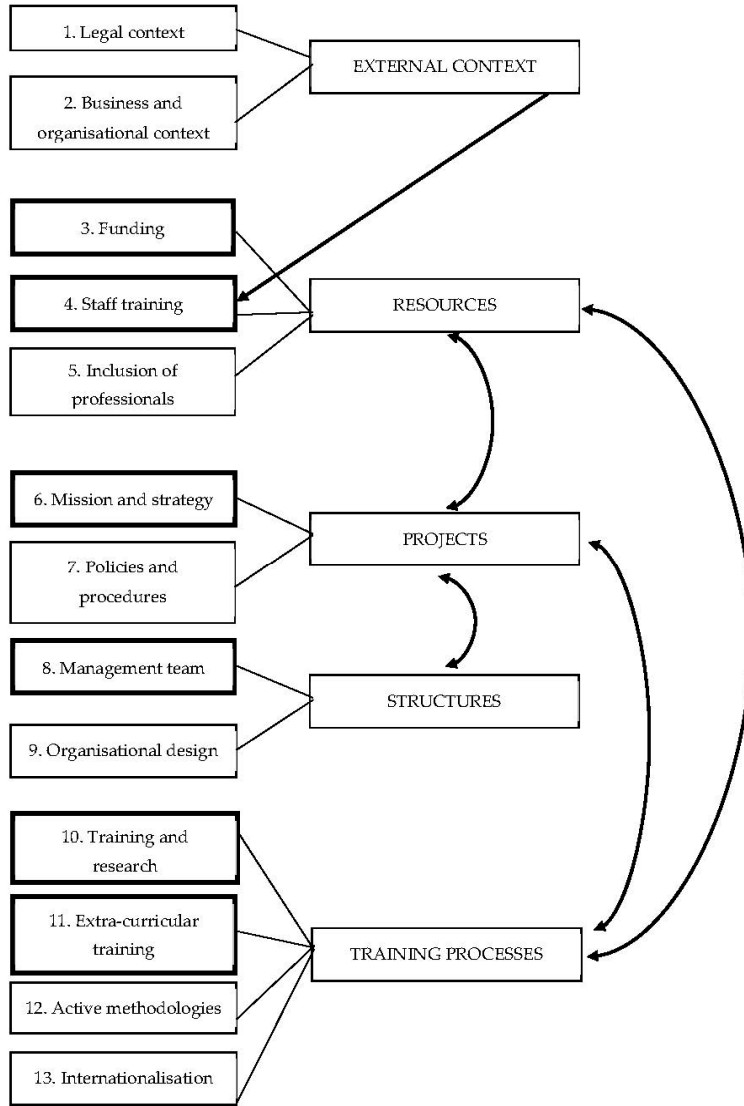

**Figure 1.** Factors of influence of the entrepreneurial university.

Contrary to what could be expected, the factors related to universities' resources seemed to be minimally related to external context factors, although they were found to be related to the entrepreneurship processes engaged in by universities. According to Hu [29], public and private funding is important for developing the entrepreneurial university, although this study did not identify a significant relationship between these two aspects. However, it was observed that resources were highly necessary for developing entrepreneurship; resources are to be understood not only in terms of financial provision, but also as human resources, including the involvement of professionals from the world of business and organisations in the design and delivery of the curriculum, and the increase in the numbers of faculty members with entrepreneurship training. It would appear that Spanish universities have compensated for the lack of external funding for entrepreneurship by utilising internal resources.

The projects related to entrepreneurship, which are crystallised into the mission, strategy, policies and procedures of universities, were found to be significantly associated with all of the entrepreneurship factors analysed. This was interesting, as it placed the documents that articulated the mission and strategy of universities in a very important position for developing entrepreneurship. Some studies have highlighted the importance of decision making in terms of entrepreneurship at strategic and organisational levels in the development of the entrepreneurial university [41]. Others have also

stressed the influence of university policies, procedures and practices on academic entrepreneurial tasks [6].

Structures, conceived as the support from the management team and the organisational design of a university, were not among the most decisive factors for developing the entrepreneurial university as a whole. This is contrary to what could have been expected, since other studies have indicated that management teams play an essential role in promoting an entrepreneurial culture [11,43]. Nevertheless, these structures were positively associated with training and research processes, which in fact seemed to be strongly related to other factors for the development of the entrepreneurial university. Consequently, universities should consider them important and pay special attention to them. This is a highly significant finding, since training and research are core objectives of Spanish universities; and according to this study these processes are strongly related to, and have a great impact on, other factors in the development of the entrepreneurial university. As these training process factors have also been proven to be the most sensitive to the influence of the other factors characterising the entrepreneurial university, acting on any of them would have an effect on the development and improvement of entrepreneurship in education. The results of this study can be a good contribution for improvement of the Spanish entrepreneurial university and its impact on the sustainable economical and social development of the region.

**Author Contributions:** Funding acquisition, M.J.B.; Project administration, M.J.B. and J.P.; Investigation, M.J.B., A.G.-O., J.P.-C. and A.A.; Conceptualization, J.P.-C. and A.A.; Methodology, M.J.B., A.G.-O., J.P.-C. and A.A.; Formal analysis, J.P.-C. and A.G.-O.; Writing—original draft preparation, A.G.-O.; Writing—review and editing, M.J.B., A.G.-O., J.P.-C. and A.A. All authors have read and agreed to the published version of the manuscript.

**Funding:** This research was funded by the Basque Government. Basic and Applied Research Projects. Project No. PI_2015_1_92. The authors acknowledge the financial support of the Department of Education, Language Policy and Culture of the Basque Government.

**Conflicts of Interest:** The authors declare no conflict of interest.

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
