# Peer review of "Developing the Entrepreneurial University: Factors of Influence"

_sustainability, doi:10.3390/su12030842_

Round 1
Reviewer 1 Report
Tnak you for the possibility to read this manuscript.
Some comments/suggestions:
"correlations" is not a keyword, in my opinion; why only external factors in : "To identify to what extent external contextual factors are related to the development of the entrepreneurial university."?, please explain; I can't find "Figure 1. Factors of influence of the entrepreneurial university."; What's the relation with sustainability? Improve the References.Author Response
RESPONSE TO REVIEWER 1 COMMENTS:
Dear reviewer 1, thank you for your commentes. Here there are the responses to the proposed suggestions:
Correlations is not a keyword: We agree. We have deleted it. "To identify to what extent external contextual factors are related to the development of the entrepreneurial university."? Please explain This objective is referred to the external factors, but the internal ones have also been taken into account in objectives 2 and 3. I can't find "Figure 1. Factors of influence of the entrepreneurial university." Figure 1 was attached in the first version but the figure, by mistake, was not available for reviewers. In this second version we have included again the figure inside the text and in .zip format. What's the relation with sustainability? Improve the References. We have added the relation between entrepreneurial university and sustainability at the end of first paragraph of the introduction (in red).Reviewer 2 Report
The title is really intriguing and it encourages the reader to look further. The abstract is well written as it includes all the essential elements: the aim, research methods and general conclusions.
The scope of research is impressive indeed, which shows the authors' commitment to the task. The list of references is adequate for the purpose.
However, the discussion section needs improvement: its first half takes the form of research conclusions rather than a real discussion. I, therefore, suggest splitting this section into research conclusion (summing up the results) and a proper discussion, with possible polemics.
Figure 1 is mentioned, but missing from my copy, even though it is a part of the research conclusion. Also, some conclusion of the factors listed would be appropriate.
I would like to see the authors' conclusion of how their research can be reflected in the Spanish Higher Education system.
Author Response
RESPONSE TO REVIEWER 2 COMMENTS:
Dear reviewer 2, thank you very much for your comments. Here there are the responses to the proposed suggestions:
The title is really intriguing and it encourages the reader to look further. The abstract is well written as it includes all the essential elements Thank you.
The scope of research is impressive indeed, which shows the authors' commitment to the task. The list of references is adequate for the purpose: Thank you very much for your comments.
However, the discussion section needs improvement: its first half takes the form of research conclusions rather than a real discussion. I, therefore, suggest splitting this section into research conclusion (summing up the results) and a proper discussion, with possible polemics. We have changed the title of the section 4, including the term “conclusion” because it includes a general conclusion of the previous results (see in red point 4. Conclusion and Discussion).
Figure 1 is mentioned, but missing from my copy, even though it is a part of the research conclusion. Also, some conclusion of the factors listed would be appropriate. Figure 1 was attached in the previous version but by mistake it was not available to the reviewers. In this second version we have included again the figure inside the text and in .zip format. We understand that this change will help to understand the conclusions.
I would like to see the authors' conclusion of how their research can be reflected in the Spanish Higher Education system. Although it was slightly mentioned before, a last sentence has been added to highlight the impact of the entrepreneurial university on the Spanish university system (see 4th paragraph of point 4 Conclusion and discussion).
Reviewer 3 Report
The authors present research based on a questionnaire answered by 84 faculty leaders on the factors that influence entrepeneurial university practices. The results are sound and merits publication but the editing and presentation needs some reformulation.
These are my comments:
1) Results: Tables 2, 3, 4, 5 and 6 should be merged. There is no need to present them in separate tables. In addition, the actual presentation of the results based on the tables, is too linear. Some of the results are presented as conclusions which is not the case. Please, make an effort of condensing the information. Please, be aware that in some factors, the categories do not appear. For example, in table 2, the category Internationalisation does not appear in F1.
2) I am confusing in the use of the term factors and the elements troughout the manuscript. And also between contextual factors and 'other factors'. Please avoid this categorization and use different terms for each case. Possible factors, elements and categories?
3) Abstract is very confusing. For example, the use of factors is confusing. To me, the introduction of influencing factors, contextual factors, internal factors is very confusing. Also, the use of the adjectives like minor influence, moderately or highly, ...., has not scientific basis. Please avoid the use of these jargon words. Also, in the abstract you need to define what are the processes. As a reader, I would appreciate what kind of processes the universities have designed to develop entrepreneurship. ... Since the abstract is too confusing from lines 15 to 25, I recommend full rewritting.
4) Objective 4, also introduces the term 'various', which is also confusing. Please substitute it by relevant information.
5) I cannot judge the full manuscript since Figure 1 is missing.
6) Discussion is good.
Author Response
RESPONSE TO REVIEWER 3 COMMENTS:
Dear reviewer 3, thank you very much for your comments. Here there are the responses to the proposed suggestions:
1) Results: Tables 2, 3, 4, 5 and 6 should be merged. There is no need to present them in separate tables. In addition, the actual presentation of the results based on the tables, is too linear. Some of the results are presented as conclusions which is not the case. Please, make an effort of condensing the information. Please, be aware that in some factors, the categories do not appear. For example, in table 2, the category Internationalisation does not appear in F1. We have merged all the tables in one (now table 2: Factors Correlation Matrix). This new table includes all existing relationships, both those that are significant and those that are not. Thank you for the comment. This idea facilitates the understanding of the results avoiding repetitions.
2) I am confusing in the use of the term factors and the elements troughout the manuscript. And also between contextual factors and 'other factors'. Please avoid this categorization and use different terms for each case. Possible factors, elements and categories? In table 1 (questionnaire), we have changed the word "items" by elements", So that "elements" has substituted the words of "sub-criteria" and "items".
We have added Stufflebeam reference to explain the institutional evaluation CIPP model that justifies the categorization of factors in contextual, input, process and product factors. In the analysis of the results, each of these categories is defined. Changes can be seen in the introduction, p. 3.
Reference [41] has been added to the list of references.
3) Abstract is very confusing. For example, the use of factors is confusing. To me, the introduction of influencing factors, contextual factors, internal factors is very confusing. Also, the use of the adjectives like minor influence, moderately or highly, ...., has not scientific basis. Please avoid the use of these jargon words. Also, in the abstract you need to define what are the processes. As a reader, I would appreciate what kind of processes the universities have designed to develop entrepreneurship. ... Since the abstract is too confusing from lines 15 to 25, I recommend full rewritting.
As explained in previous comment, we expect the categories for analysis (CIPP model) have been clearly understood.
Adjectives minor, moderately .... have been used according to classification made by Bisquerra in the field of educational sciences, Bisquerra distinguishes correlations with too high, high, moderate, low and too low significance.
> 0.80 |
Too high |
0.60-0.79 |
High |
0.40-0.59 |
Moderate |
0.20-0.39 |
Low |
<0.20 |
Too low |
4) Objective 4, also introduces the term 'various', which is also confusing. Please substitute it by relevant information. We understand you refer to objective 3. We have deleted the word "various" and substituted it by "relevant".
5) I cannot judge the full manuscript since Figure 1 is missing.
Figure 1 was attached in the previous version but by mistake it was not available to the reviewers. In this second version we have included again the figure inside the text and in .zip format.
6) Discussion is good. Thank you for your comment.
Round 2
Reviewer 2 Report
Thank you for your reaction to my suggestions. The paper has been improved but I still feel some more changes are necessary. I appreciate the addition to the title of chapter 4, yet I would still insist on its division into two separate parts: research conclusions and final conclusions & discussions. Such change would make your message clearer and it would be less confusing for the reader.
Figure 1 does explain a lot but it should be introduced and described in more than just one sentence.
Author Response
RESPONSE TO REVIEWER 2 COMMENTS:
Thank you for your reaction to my suggestions. The paper has been improved but I still feel some more changes are necessary. I appreciate the addition to the title of chapter 4, yet I would still insist on its division into two separate parts: research conclusions and final conclusions & discussions. Such change would make your message clearer and it would be less confusing for the reader.
Thank you very much for your suggestions. We have divided the chapter 4 into two parts: point 4: Conclusions and point 5 Discussion.
Figure 1 does explain a lot but it should be introduced and described in more than just one sentence.
Figure 1 has been included in point 4: Conclusions and an explanation of it has been added (in blue).
Reviewer 3 Report
The authors have commented the main points, but I have still some minor comments that should be addressed:
1) The authors do not need to be listed as 1, 2, 3 and 4, since they are in the same location.
2) Table 1. Since the journal is internationaly oriented, please DO NOT use commas. You should be presenting points.
3) Abstract: There are two phrases that are still confusing: a) seemed to be strongly related to other essential factors (please, describe what are the other factors), and b) show the existing (the word existing is too confusing).
4) Line 121, Delete 'As said before'.
Author Response
RESPONSE TO REVIEWER 3 COMMENTS:
The authors have commented the main points, but I have still some minor comments that should be addressed:
The authors do not need to be listed as 1, 2, 3 and 4, since they are in the same location.We have modified this aspect, although we had understood, following the template, that they should have been listed.
María José Bezanilla *, Ana García-Olalla, Jessica Paños-Castro and Arantza Arruti 1
1 University of Deusto; marijose.bezanilla@deusto.es; ana.garciaolalla@deusto.es; jessicapanos@deusto.es; aarruti@deusto.es
Table 1. Since the journal is internationaly oriented, please DO NOT use commas. You should be presenting points.
Done. We assume you refer to Table 2.
Abstract: There are two phrases that are still confusing: a) seemed to be strongly related to other essential factors (please, describe what are the other factors), and b) show the existing (the word existing is too confusing).We have modified the sentence related to other essential factors, highlighting the most significant one. (see changes in blue)
We have deleted the word “existing”, hoping this makes it more clear.
4) Line 121, Delete 'As said before'.
Done
Round 3
Reviewer 2 Report
The authors have improved the text, thus greatly enhancing its value.
Author Response
Thanks very much for your comments !
Kind regards,
Marijose Bezanilla